# CXCL8 Up-Regulated LSECtin through AKT Signal and Correlates with the Immune Microenvironment Modulation in Colon Cancer

**DOI:** 10.3390/cancers14215300

**Published:** 2022-10-28

**Authors:** Shaojun Fang, Xianshuo Cheng, Tao Shen, Jian Dong, Yunfeng Li, Zhenhui Li, Linghan Tian, Yangwei Zhang, Xueyan Pan, Zhengfeng Yin, Zhibin Yang

**Affiliations:** 1Department of Colorectal Surgery, The Third Affiliated Hospital of Kunming Medical University/Yunnan Cancer Hospital, Kunming 650118, China; 2Department of Radiology, The Third Affiliated Hospital of Kunming Medical University/Yunnan Cancer Hospital, Kunming 650118, China; 3Department of Yunnan Tumor Research Institute, The Third Affiliated Hospital of Kunming Medical University/Yunnan Cancer Hospital, Kunming 650118, China; 4Molecular Oncology Laboratory, Eastern Hepatobiliary Surgery Hospital, Second Military Medical University, Shanghai 200433, China

**Keywords:** CXCL8, LSECtin, immune microenvironment, colon cancer, liver metastasis

## Abstract

**Simple Summary:**

Patients with high expression of CXCL8 are not sensitive to immune checkpoint inhibitors (ICIs) treatment, but the mechanism is unclear. LSECtin is the immune checkpoint ligand of LAG3, and is considered as an important factor of ICIs resistance. This study confirmed the role of CXCL8 and LSECtin in immune microenvironment modulation of colon cancer. The expression of CXCL8 is positively correlated with more than 40 immune checkpoints. CXCL8 could up-regulate LSECtin through AKT signal and promoted the proliferation and invasion ability of colon cancer. These results may be important reasons for the primary drug resistance of ICIs in colon cancer.

**Abstract:**

Background: The role of CXCL8 and LSECtin in colon cancer liver metastasis and immune checkpoint inhibitors (ICIs) treatment effect were widely recognized. However, the regulatory role of CXCL8 on LSECtin is still unclear. Methods: The expression of CXCL8 or LSECtin was analyzed by TCGA database, and verified by GES110225 and clinical samples. The relationship between the expression of CXCL8 or LSECtin and immune cells infiltration, Kyoto Encyclopedia of Genes and Genomes (KEGG) pathway, Gene Ontology (GO) items, stromal score, Estimation of STromal and Immune cells in MAlignant Tumours (ESTIMAT) immune score, tumor mutation burden (TMB), mismatch repair gene and immune checkpoints expression were analyzed by Spearman. The effects of CXCL8 on LSECtin expression, proliferation, and invasion ability were clarified by recombinant CXCL8 or CXCL8 interfering RNA. Results: In colon cancer, the expression of CXCL8 was higher, but LSECtin was lower than that in normal mucosa. The expression of CXCL8 or LSECtin was significantly positively correlated with immune cells infiltration, stromal score, ESTIMATE immune score, TMB, and immune checkpoints expression. The expression of LSECtin was closely related to the cytokine-cytokine receptor interaction pathway and response of chemokine function, such as CXCL8/CXCR1/2 pathway. There was a significant positive correlation between the expression of CXCL8 and LSECtin in colon cancer. CXCL8 up-regulated LSECtin through AKT signal and promoted the proliferation and invasion ability of colon cancer. Conclusions: CXCL8 up-regulated LSECtin by activating AKT signal and correlated with the immune microenvironment modulation in colon cancer.

## 1. Introduction

Immune checkpoints are a group of molecules expressed mainly on immune and tumor cells that can function as inhibitory mediators for immune escape in tumor. Immune checkpoint inhibitors (ICIs) can re-activate anti-tumor immune response and inhibit tumor growth, and have become the focus of tumor therapy. At present, ICIs targeting PD-1/PD-L1 or CTLA4 have been rapidly developed. Treatment with ICIs have improved the survival in a variety of cancer. However, not all patients can benefit from ICIs [1]. Almost 90% of colorectal cancer (CRC) patients show primary resistance to anti-PD-1/PD-L1 or anti-CTLA4 [2,3]. Thus, understanding the mechanisms of ICIs primary resistance, as well as identifying novel and effective biomarkers, is crucial for improving the sensitivity of ICIs.

Chemokine ligand 8 (CXCL8), also known as interleukin-8 (IL-8), plays a biological role by binding to CXCR1 and CXCR2. CXCL8 is an indispensable important inflammatory response factor and immunosuppressive factor in the tumor microenvironment and has been shown to be up-regulated in a variety of tumor tissues or tumor patient’s serum [4]. Our previous studies suggested that CXCL8 could promote liver metastasis by inducing epithelial mesenchymal transformation (EMT), promoting anoikis resistance and other mechanisms. High expression of CXCL8 in cancer cells suggests poor survival prognosis in colorectal cancer [5,6]. Latest evidence suggested that CXCL8 has the ability to recruit immunosuppressive cells and inhibit anti-tumor immune response [7]. Data from vitro and clinical studies also suggested that the activation of CXCL8/CXCR1/2 signal axis is an important factor in regulating the immune microenvironment of colon cancer and leading to less benefit of ICIs treatment in colon cancer [8,9,10]. Therefore, CXCL8 is a key factor of ICIs primary resistance in colon cancer. However, the mechanism is still unclear.

Liver sinusoidal endothelial cell lectin (LSECtin), a type II membrane protein encoded by CLEC4G gene, belongs to the C-type lectin receptor family [11]. Available evidence suggested that LSECtin is an immune checkpoint ligand for lymphocyte activation gene 3 (LAG-3). Studies have revealed that the interaction between LSECtin and LAG-3 can inhibit the secretion of IFN-γ by T cells, and promote melanoma cell proliferation and immune escape. LAG-3 antibody C9B7W could block the binding of LAG-3 to the LSECtin, inhibit the proliferation of effector T cells and the growth of transplanted tumor [12]. It has been determined that blocking LAG3/LSECtin signal can not only restore the cytotoxic activity of T lymphocytes, but also reduce the inhibitory function of regulatory T (Treg) cells. It is noteworthy that LSECtin can significantly enhance the adhesion, migration, and invasion of colon cancer cells [13]. Results from Na et al. [13] showed that the probability of liver metastasis in LSECtin knockout group was significantly lower than that in wild-type group, the level of serum LSECtin in patients with liver metastasis was significantly higher than that in patients without liver metastasis. Therefore, LSECtin is an important immune checkpoint ligand to promote liver metastasis of colon cancer, immune escape is a potential mechanism.

In this study, we analyzed the correlation between CXCL8 and LSECtin through the cancer genome atlas (TCGA) and gene expression omnibus (GEO) public data resources and explored the potential role of the CXCL8 or LSECtin in the immune microenvironment modulation of colon cancer. Later, we demonstrated that CXCL8 up-regulated LSECtin expression by activating AKT signal in colon cancer cells. Our results provide a theoretical basis for clarifying the mechanism of primary drug resistance of ICIs, and might provide a new idea for solving the primary drug resistance of ICIs in colon cancer.

## 2. Materials and Methods

### 2.1. Data Source

The different expression of CXCL8 or LSECtin between cancer tissues and normal mucosa tissues of 20 tumor types were analyzed by TCGA. Combined with Genotype-Tissue Expression (GTEx) database, different expression of CXCL8 or LSECtin in 27 tumor types were also compared. GSE110225 data were download from the GEO (https://www.ncbi.nlm.nih.gov/geo/ (accessed on 12 May 2022) for verification.

### 2.2. Immune Correlation Analyst

Tumor IMmune Estimation Resource (TIMER) (https://cistrome.shinyapps.io/timer/ (accessed on 12 May 2022)) is an open online tool for systematic analysis of immune cell infiltration in different cancer types. The relationship between the expression of CXCL8 or LSECtin with the infiltration of six types of immune cells in colon cancer such as B cells, CD4+ T cells, CD8+ T cells, neutrophils, macrophages, and dendritic cells were analyzed by TIMER. The correlation between the expression of CXCL8 or LSECtin and stromal score, Estimation of STromal and Immune cells in MAlignant Tumours (ESTIMAT) immune score, tumor mutation burden (TMB), microsatellite instability (MSI), five mismatch repair genes mutation (MLH1, MSH2, MSH6, PMS2, EpCAM), and more than 40 immune checkpoints expression were analyzed by Spearman correlation analysis.

### 2.3. KEGG and GO Functional Enrichment Analysis

Linkedomics (http://www.linkedomics.org/login.php (accessed on 12 May 2022)) is a public platform for analyzing and comparing different types of cancer multiomics data. Kyoto Encyclopedia of Genes and Genomes (KEGG) pathways and Gene Ontology (GO) terms of CXCL8 and LSECtin were enriched according to the standard of FDR < 0.05 and 1000 simulations through the function module of Linkedomics.

### 2.4. Human Colon Cancer Tissues

Ten pairs of colon cancer tissues and their corresponding normal mucosa tissues were obtained from Yunnan Cancer Hospital (The Third Affiliated Hospital of Kunming Medical University) and used in accordance with the policies of the institutional review board of the hospital (NO. KYCS2022072). All tissue diagnoses were confirmed histologically. Informed consent was obtained from all subjects involved in the study.

### 2.5. Cell Lines and Cell Culture

The human colon cancer cell lines SW480 and SW620 (American Type Tissue Culture Collection, Manassas, VA, USA) were maintained in RPMI 1640 (Invitrogen, Waltham, MA, USA) medium supplemented with 10% fetal bovine serum (FBS) (Invitrogen, Waltham, MA, USA), streptomycin (100 mg/mL), and penicillin (100 mg/mL) in a humidified incubator containing 5% CO_2_ at 37 °C. Human recombinant CXCL8 (100 ng/mL) (R&D Systems, Minneapolis, MN, USA) and AKT inhibitor MK2206 (Selleck Chemicals, Houston, TX, USA) were used for cell viability analysis.

### 2.6. RNA Interference

CXCL8 knockdown in colon cancer cells was performed by siRNA transfection. According to the scheme of siRNA design company (Genepharma, Suzhou, China), CXCL8 siRNA, non-specific siRNA, and Lipofectamine 3000 (Invitrogen, Waltham, MA, USA) were diluted respectively, mixed gently, and incubated at room temperature for 20 min. siRNA Lipofectamine 2000 mixture was added to colon cancer cells grown in 6-well plates to achieve 80–90% confluence. The transfection reagent was changed after 6 h. Non-specific siRNA was used as normal control. Information about siRNA is described below.


**CXCL8 siRNA1**
CXCL-8homo-215 GGUGCAGUUUUGCCAAGGATT         UCCUUGGCAAAACUGCACCTT
**CXCL8 siRNA2**
CXCL-8homo-430 GAAGAGGGCUGAGAAUUCATT         UGAAUUCUCAGCCCUCUUCTT
**CXCL8 siRNA3**
CXCL-8homo-557 GCCAGAUGCAAUACAAGAUTT         AUCUUGUAUUGCAUCUGGCTT
**Non-specific siRNA**
Negative control sense 5′UUCUCCGAACGUGUCACGUTT-3Antisense      5′-ACGUGACACGUUCGGAGAATT-3′

### 2.7. Proliferation and Invasion Assays

MTT was used to evaluate the proliferation of colon cancer cells. Cells treated with CXCL8 or transfected with CXCL8 siRNA/non-specific siRNA were seeded into 96 well plates (2000 cells/well). MTT (20 μL/well) was added to each plate at 0, 24, 48, 72 h, and incubated for 4 h in a cell incubator at 37 °C and 5% CO_2_, dissolved the crystallize by dimethyl sulfoxide (DMSO), and then detected the absorbance at 450 nm in the microplate reader. Transwell assay was used to evaluate the invasive ability of colon cancer cells after treating with CXCL8 or interference with CXCL8 siRNA. Matrigel (BD Biosciences, Haryana, India) was used to coat the basement membrane of Transwell chamber. Cells were diluted with serum-free medium and seeded into Transwell chamber (5000 cells/chamber). The lower chamber was filled with serum containing medium or CXCL8. The chamber was taken out, stained with crystal violet after 48 h, and 5 points were randomly taken under 200 times microscope for photographing and counting.

### 2.8. Western Blot

After treated with CXCL8 or transfected with CXCL8 siRNA/non-specific siRNA for 48 h, the cells were lysed by RIPA (Beyotime, Haimen, China) supplemented with protease inhibitor (Beyotime, Haimen, China). Electrophoresis was carried out on sodium dodecyl sulfate polyacrylamide gel according to the standard of 30 ug/lane. After electrophoresis, the protein was transferred to polyvinylidene fluoride membrane. The primary antibody was incubated overnight at 4 °C and the secondary antibody labeled with horseradish peroxidase was incubated at room temperature for 1 h. Western Blot bands were detected by emitter coupled logic (ECL) solution (Beyotime, Haimen, China). The image was semi quantitatively analyzed by Image J software.

Primary antibodies were as follows: rabbit anti-human p-AKT (Ser473) (Cell Signaling Technology, Danvers, MA, USA), rabbit anti-human AKT (C67E7) (Cell Signaling Technology, Danvers, MA, USA), and anti-human LSECtin (ab181196) (Abcam, Cambridge, UK). Primary antibodies were as follows: Rabbit anti human p-AKT (ser473) (cell signaling technology, Danvers, MA, USA), Rabbit anti human AKT (c67e7) (cell signaling technology, Danvers, MA, USA), and Rabbit anti human LSECtin (ab181196) (Abcam).

### 2.9. RT-qPCR

Total RNA was extracted from ten pairs of colon cancer tissues and normal mucosa pretreated with Trizol (servicebio, Wuhan, China). After the concentration and purity of RNA were detected, the RNA was reverse transcribed with cDNA synthesis Kit (servicebio, Wuhan, China). Refer to the instructions of the qPCR Kit (servicebio, Wuhan, China), qPCR was performed under the thermal cycle conditions: 40 cycles, 95 °C for 60 s, 95 °C for 20 s, 55 °C for 20 s and 72 °C for 30 s. The gene expression was calculated by 2^−∆∆Ct^ method. The primers are given in Table 1.

### 2.10. Statistical Analysis

The data are expressed as mean ± standard deviation. The different expression of CXCL8 and LSECtin in cancer tissues and normal mucosal tissues were compared by paired *t*-test. The correlation between CXCL8 and LSECtin was analyzed by Spearman correlation analysis. The comparison of proliferation and invasion ability was statistically analyzed by one-way analysis of variance (ANOVA). *p* < 0.05 means the difference is statistically significant.

## 3. Results

### 3.1. Higher Expression of CXCL8 Is Related to the Activation of Immune Related Function in Colon Cancer

We evaluated the transcriptional level of CXCL8 in 20 tumor types through TCGA database. The results showed that the expression of CXCL8 in colon cancer was significantly higher than that in normal mucosa (Figure 1A). These results were verified by TCGA and GTEX integrated data and GSE110225 data (Figure 1B,C). In addition, we detected CXCL8 from 10 pairs of colon cancer tissues and adjacent normal mucosa by RT-qPCR and found the similar results (t = 3.312, *p* = 0.0091) (Figure 1D).

Immune cell infiltration analysis showed that the expression of CXCL8 was significantly positively correlated with the infiltration of CD8+ T lymphocytes, DC cells, macrophages and neutrophils (Figure 2A). GSEA function enrichment suggested that the immune related KEGG pathways (Figure 2B) and GO terms (Figure 2C) were significantly enriched in patients with high CXCL8 expression, such as KEGG pathways of cytokine–cytokine receptor interaction, chemokine signaling pathway, intestinal immune network for IgA production and Toll-like receptor signaling pathway, GO terms of macrophage activation and leukocyte migration.

### 3.2. Higher Expression of CXCL8 Is Positively Related to ICIs Efficacy Markers in Colon Cancer

Next, we analyzed the correlation between CXCL8 and ICIs efficacy markers. The results showed that CXCL8 expression in colon cancer tissues was significantly positively correlated with stromal score (r = 0.416, *p* < 0.0001), ESTIMATE immune score (r = 0.465, *p* < 0.001) (Figure 3A,B), tumor mutation burden (TMB) (*p* = 0.00057) (Figure 3C) and microsatellite instability (MSI) (*p* = 1.4 × 10^−5^) (Figure 3D), and negative correlation with mismatch repair protein EpCAM mutation (*p* < 0.05) (Figure 3E). Moreover, CXCL8 expression was significantly positively correlated with the expression of CTLA4, CD86, CD274, CD276, TIGIT, VSIR, LAG3, HAVCR2, PDCD1 and other immune checkpoints (*p* < 0.05) (Figure 3F).

### 3.3. Higher Expression of LSECtin Is Related to the CXCL8/CXCR1/2 Axis Related Function in Colon Cancer

Referring to CXCL8 analysis methods, we found that all of the results from TCGA data, TCGA and GTEX integrated data and GSE110225 data indicated that the expression of LSECtin in colon cancer tissues was significantly lower than that in normal mucosa tissues (Figure 4A–C). These results were also verified by 10 pairs of clinical paired samples (t = 13.30, *p* < 0.0001) (Figure 4D).

Immune cell infiltration analysis showed that the expression of LSECtin was significantly positively correlated with the infiltration of 6 types of immune cells, such as B cells, CD4+ T lymphocytes, CD8+ T lymphocytes, DC cells, macrophages, and neutrophils (Figure 5A). GSEA function enrichment suggested that the immune-related KEGG pathway (Figure 5B) and GO terms (Figure 5C) were significantly enriched in patients with high expression of LSECtin, such as KEGG pathways of cytokine–cytokine receptor interaction, intestinal immune network for IgA production and allograft rejection, GO terms of response to chemokine, inflammatory response to antigen stimulus and myeloid dendritic cell activation. CXCL8/CXCR1/2 signal axis is the key pathway in the cytokine–cytokine receptor interaction (Figure 5D) and response to chemokine (Figure 5E) function.

### 3.4. Higher Expression of LSECtin Is Positively Related to ICIs Efficacy Markers in Colon Cancer

Similarly, we further analyzed the correlation between LSECtin and ICIs efficacy markers, such as TMB, stromal score, ESTIMAT immune score, MSI and mismatch repair gene mutation. The results showed that LSECtin expression in colon cancer tissues was significantly positively correlated with stromal score (r = 0.547, *p* = 3.66 × 10^−37^), ESTIMAT immune score (r = 0.608, *p* = 1.1 × 10^−42^) (Figure 6A,B), TMB (*p* = 0.042) (Figure 6C), MSI (*p* = 0.029) (Figure 6D), and negatively correlated with mismatch repair gene MSH2 and EpCAM mutations (*p* < 0.05) (Figure 6E). LSECtin expression was significantly positively correlated with the expression of CTLA4, CD86, CD274, CD276, TIGIT, VSIR, LAG3, HAVCR2, PDCD1, and other immune checkpoints (*p* < 0.05) (Figure 6F).

### 3.5. The Expression of CXCL8 Is Positively Correlated with LSECtin in Colon Cancer

In order to clarify the correlation between CXCL8 and LSECtin in colon cancer, we performed the analysis in the TIMER database and found that there was a significant positive correlation between CXCL8 and LSECtin expression (r = 0.357, *p* = 3.24 × 10^−15^) (Figure 7A). This result was also verified in the clinical paired sample test (r = 0.9578, *p* = 3.24 × 10^−15^) (Figure 7B).

### 3.6. CXCL8 Regulates the Expression of LSECtin through AKT Signal in Colon Cancer

To clarify the expression regulation relationship between CXCL8 and LSECtin, we designed three CXCL8 interfering RNAs. We found that CXCL8 RNAi-2 had the best inhibitory effect on CXCL8 expression (Figure 7C). Therefore, we chose CXCL8 RNAi-2 to complete the relevant experiments. We performed cell proliferation and invasion experiments by recombinant human CXCL8 (100 ng/mL) and CXCL8 RNAi-2. Recombinant human CXCL8 can significantly promote the proliferation (Figure 7D) and invasion of colon cancer cells (Figure 7F,G). CXCL8 RNAi-2 can significantly inhibit the proliferation (Figure 7E) and invasion of colon cancer cells (Figure 7H,I). Through WB detection, it was found that recombinant human CXCL8 could significantly activate AKT signal, accompanied by up-regulation of LSECtin (Figure 8A–C), and CXCL8 RNAi-2 could significantly inhibit AKT signal, accompanied by down-regulation of LSECtin (Figure 8D–F). After the AKT signal was blocked by MK2206, we found that while the role of CXCL8 in promoting the invasion of colon cancer cells was inhibited (Figure 8A–C), the regulatory effect of CXCL8 on LSECtin also disappeared (Figure 8D–F).

## 4. Discussion

Emerging clinical evidence suggests that CXCL8/CXCR1/2 signal axis plays a key role in immune escape and CXCL8 high expressed colon cancer seldom benefit from ICIs therapy [14,15]. However, the mechanism is still unclear. Up-regulation of alternate immune checkpoints is one of the important mechanisms for ICIs primary resistance [16,17]. LSECtin, a ligand of immune checkpoint LAG3, has been shown to be an important factor in promoting liver metastasis of colon cancer [18,19]. Our results showed that the expression of LSECtin was closely related to the activation of CXCL8/CXCR1/2 signal axis in colon cancer. Moreover, our previous studies and existing evidence suggested that CXCL8/CXCR1/2 is an important signal to promote liver metastasis of colon cancer [4,6]. Various evidence suggested that CXCL8 may be an important factor in regulating the expression of LSECtin. In order to validate this hypothesis, we carried out a preliminary exploration and found that CXCL8 could up-regulate LSECtin expression through AKT signal. CXCL8/AKT/LSECtin activation is positively related to the immune microenvironment regulation, and might be an important mechanism of ICIs primary drug resistance in colon cancer.

First, we analyzed the different expression of CXCL8 and LSECtin in colon cancer and the main biological functions involved. The expression of CXCL8 in cancer tissues was significantly higher than that in normal mucosal tissues, which was consistent with our previous results [5,6]. GSEA function enrichment analysis suggested that high expression of CXCL8 is mainly involved in the immune related KEGG pathways and GO terms, such as KEGG pathways of cytokine-cytokine receptor interaction, chemokine signaling pathway, intestinal immune network for IgA production and Toll-like receptor signaling pathway, GO terms of macrophage activation and leukocyte migration. The expression of LSECtin in cancer tissues is significantly lower than that in normal mucosal tissues. Its high expression is also mainly involved in the immune related KEGG pathway and GO terms, such as KEGG pathways of cytokine-cytokine receptor interaction, intestinal immune network for IgA production and allograft rejection, GO terms of response to chemokine, inflammatory response to antigen stimulus and myeloid dendritic cell activation.

Previous studies have shown that cytokine–cytokine receptor interaction and chemokine signaling pathway are important mechanisms in tumor immune microenvironment regulation, immune escape, and immunotherapy tolerance [20,21,22]. CXCL8/CXCR1/2 is one of the key signals in these two pathways. Studies have shown that CXCL8 is closely related to the immune cells infiltration in colon cancer [22]. CXCL8 is a key chemokine of the innate immune system that recruits immunosuppressive cells, such as neutrophils, myeloid derived suppressor cells (MDSC) and monocytes [23]. Our results showed that CXCL8 expression is positively correlated with LSECtin expression in colon cancer. Therefore, CXCL8/CXCR1/2 signal may regulate the expression of LSECtin and play an important role in the immune microenvironment regulation in colon cancer.

To clarify the role of CXCL8 or LSECtin in tumor immune microenvironment and ICIs therapy, we carried out further analysis on immune cells infiltration and ICIs markers. The results suggest that the expression of CXCL8 or LSECtin in colon cancer is significantly positively related with the infiltration of CD8+ T lymphocytes, DC cells, macrophages, and neutrophils, which is a key factor affecting the therapeutic efficacy of ICIs [24]. Our results are consistent with those reported by Yang et al. [22]. CXCL8 can recruit neutrophils and MDSC around cancer cells to form neutrophil extracellular traps (NETs) and protect them from cytotoxicity mediated by CD8+ T cells and natural kill (NK) cells, which in turn affects the efficacy of ICIs therapy [25]. Zhang et al. [10]. suggested that TNF-γ can inhibit the expression of CXCL8, inhibit the recruitment of CXCR2+, CD68+ macrophages, and increase the sensitivity of pancreatic cancer against anti-PD-1. Five common immune checkpoint efficacy predict markers [26,27], such as TMB, stromal score, ESTIMAT immune score, MSI and mismatch repair gene mutation were analyzed, and the results showed that the expression of CXCL8 or LSECtin is positively correlated with these markers. Therefore, patients with high expression of CXCL8 or LSECtin may benefit from ICIs.

However, the existing clinical data suggest that the high expression of CXCL8 in serum portends that patients cannot benefit from anti-PD-1/PD-L1 or anti-CTLA4 treatment. Regulatory immune cell infiltration, disruption of antigen presentation mechanism, immunosuppressive microenvironment, and cancer stem cells are the main mechanisms of primary drug resistance of ICIs, up-regulation of other immune checkpoints is one of the important mechanisms [17]. We analyzed the relationship between the expression of CXCL8 or LSECtin with more than 40 immune checkpoints. The results suggest that patients with high expression of CXCL8 or LSECtin are often accompanied by the high expression of immune checkpoints such as CTLA4, CD86, CD274, CD276, TIGIT, VSIR, LAG3, HAVCR2, and PDCD1. This evidence indirectly explains the phenomenon that patients with high CXCL8 expression cannot benefit from anti-PD-1/PD-L1 or anti-CTLA4. So, the increased expression of immune checkpoints may be the main reason in weakening the sensitivity to anti-PD-1/PD-L1 or anti-CTLA4 treatment in patients with CXCL8 overexpression.

In view of the close correlation between the expression of CXCL8 and LSECtin as described above, we added human recombinant CXCL8 or knocked down the expression of CXCL8 in SW480 and SW620 cells. Later, we detected the proliferation ability, invasion ability, and the expression of LSECtin. Results showed that human recombinant CXCL8 enhanced the up-regulation of LSECtin, the proliferation and invasion ability of SW480 and SW620 cells by activating the AKT signaling pathway. After inhibiting AKT signal with MK2206, the LSECtin expression, and the proliferation and invasion ability of SW480 and SW620 cells were significantly inhibited. After interfering with the expression of CXCL8 in SW480 and SW620 cells, we observed the suppression of AKT signal, the down-regulation of LSECtin, and the inhibition of cell proliferation and invasion ability. Therefore, CXCL8 can promote the expression of LSECtin through AKT signal activation. These results partly explain the phenomenon that patients with high CXCL8 expression cannot benefit from anti-PD-1/PD-L1 or anti-CTLA4 treatment.

This study evaluated the role of CXCL8 and LSECtin in the immune microenvironment and ICIs efficacy prediction of colon cancer, and clarified the regulatory mechanism of CXCL8 regulating LSECtin through AKT activation. This mechanism partly explains the phenomenon of primary drug resistance to anti-PD-1/PD-L1 or anti-CTLA4 in colon cancer patients with high expression of CXCL8. Our research combines bioinformatics and basic experimental verification. However, whether blocking CXCL8/CXCR1/2 signal or LAG3/LSECtin can increase the efficacy of anti-PD-1/PD-L1 or anti-CTLA4 in patients with colon cancer is still lack of effective experimental verification. At the same time, it is still unknown whether CXCL8 regulates the other immune checkpoints expression. This field is also worth exploring.

## 5. Conclusions

In conclusion, this study is the first to clarify the regulatory effect of CXCL8 on LSECtin, which is an important mechanism for CXCL8 to regulate the immune microenvironment of colon cancer. The results could provide a theoretical basis for clarifying the mechanism of anti-PD-1/PD-L1 or anti-CTLA4 primary drug resistance in colon cancer, and provide a new idea for solving the phenomenon of anti-PD-1/PD-L1 or anti-CTLA4 primary drug resistance in colon cancer.

## Figures and Tables

**Figure 1 cancers-14-05300-f001:**
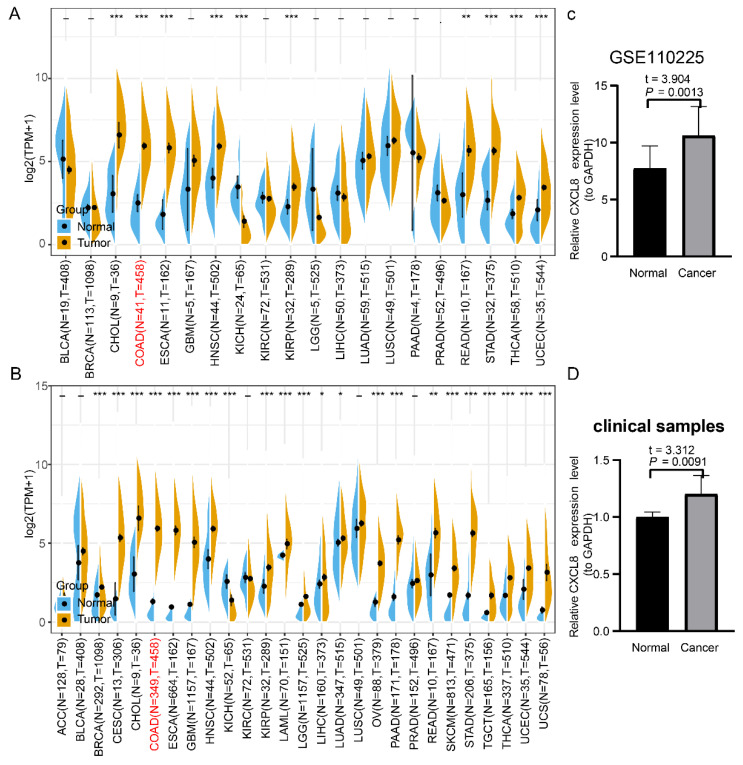
Expression level of CXCL8 in colon cancer. (**A**) The expression level of CXCL8 in different tumor types (TCGA database). (**B**) The expression level of CXCL8 in different tumor types (TCGA + GTEX database). (**C**) Expression level of CXCL8 in colon cancer and normal mucosa (GSE110225). (**D**) Expression level of CXCL8 in colon cancer and normal mucosa (clinical samples). The caption for the red font is colon adenocarcinoma (COAD). * *p* < 0.05, ** *p* < 0.01, *** *p* < 0.001.

**Figure 2 cancers-14-05300-f002:**
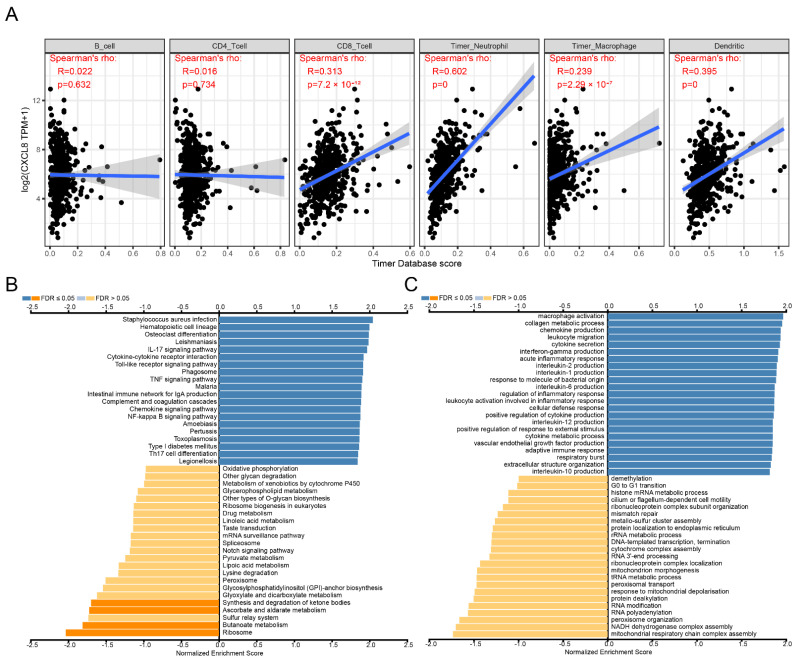
Identification of biological function characteristics of CXCL8 in colon cancer. (**A**) Relationship between CXCL8 and immune cell infiltration. (**B**) KEGG pathway enrichment analysis. (**C**) GO items enrichment analysis.

**Figure 3 cancers-14-05300-f003:**
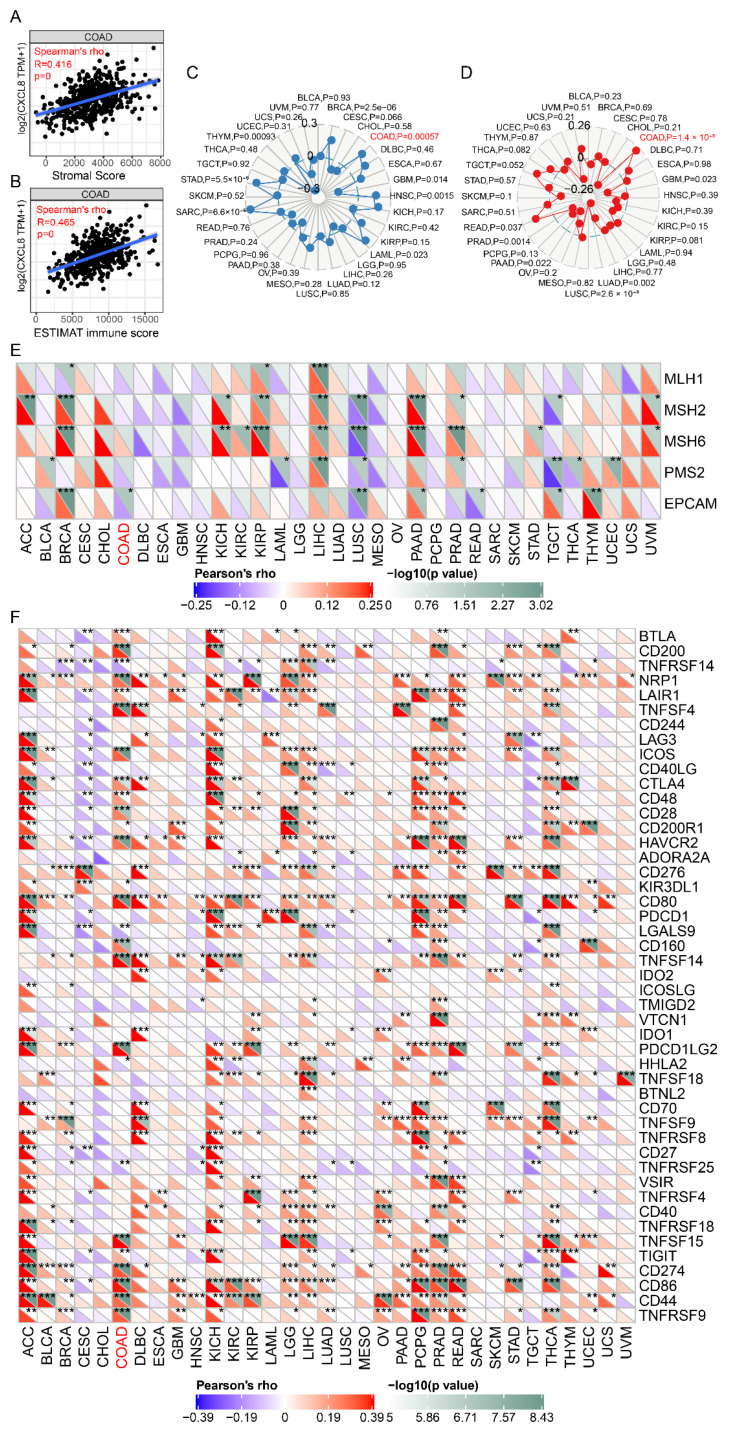
Relationship between CXCL8 and stromal score, ESTIMAT immune score, TMB, MSI, mismatch repair gene and immune checkpoints. (**A**) Relationship between CXCL8 and stromal score. (**B**) Relationship between CXCL8 and ESTIMAT immune score. (**C**) Relationship between CXCL8 and TMB. (**D**) Relationship between CXCL8 and MSI. (**E**) Relationship between CXCL8 and mismatch repair gene. (**F**) Relationship between CXCL8 and immune checkpoints. The caption for the red font is COAD. * *p* < 0.05, ** *p* < 0.01, *** *p* < 0.001.

**Figure 4 cancers-14-05300-f004:**
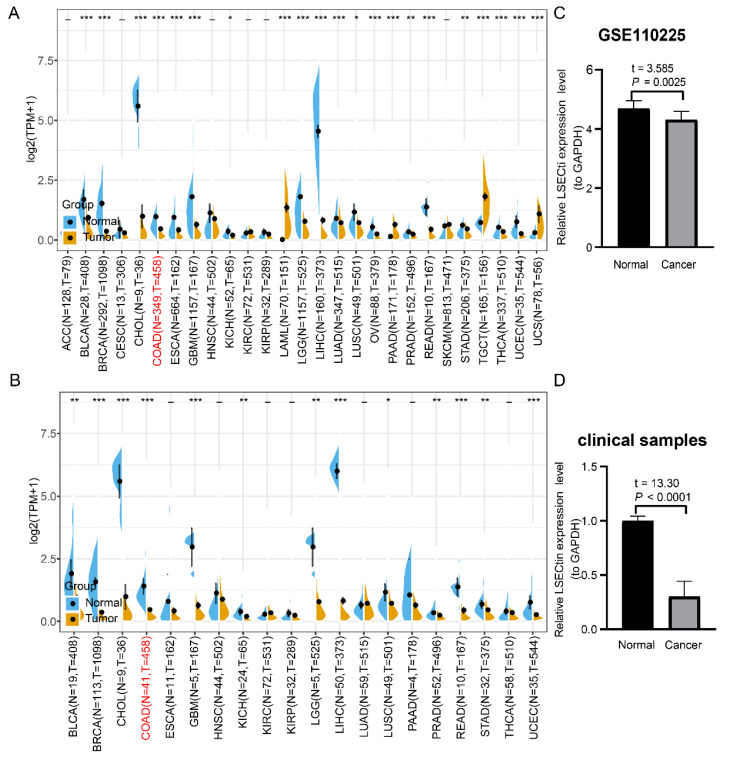
Expression level of LSECtin in colon cancer. (**A**) The expression level of LSECtin in different tumor types (TCGA database). (**B**) The expression level of LSECtin in different tumor types (TCGA + GTEX database). (**C**) Expression level of LSECtin in colon cancer and normal mucosa (GSE110225). (**D**) Expression level of LSECtin in colon cancer and normal mucosa (clinical samples). The caption for the red font is COAD. * *p* < 0.05, ** *p* < 0.01, *** *p* < 0.001.

**Figure 5 cancers-14-05300-f005:**
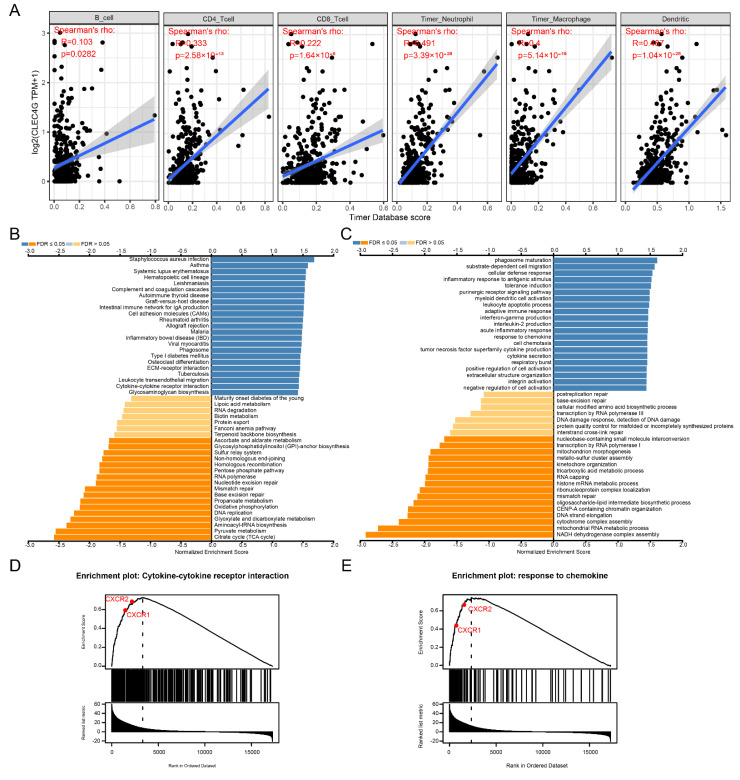
Identification of biological function characteristics of LSECtin in colon cancer. (**A**) Relationship between LSECtin and immune cell infiltration. (**B**) KEGG pathway enrichment analysis. (**C**) GO items enrichment analysis. (**D**) Enrichment diagram of cytokine-cytokine receptor interaction pathway. (**E**) Response to chemokine function enrichment diagram.

**Figure 6 cancers-14-05300-f006:**
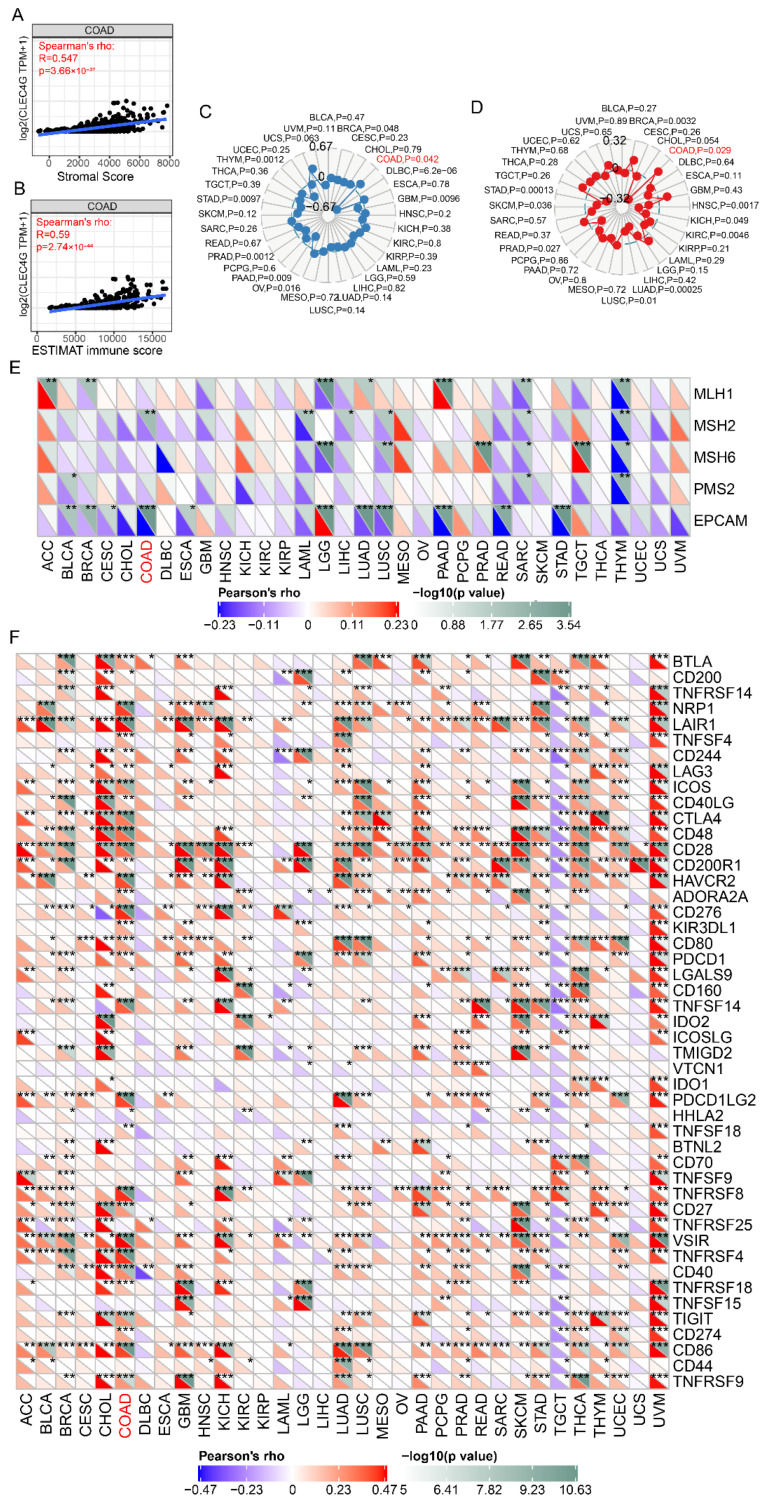
Relationship between LSECtin and stromal score, ESTIMAT immune score, TMB, MSI, mismatch repair gene and immune checkpoints. (**A**) Relationship between LSECtin and stromal score. (**B**) Relationship between LSECtin and ESTIMAT immune score. (**C**) Relationship between LSECtin and TMB. (**D**) Relationship between LSECtin and MSI. (**E**) Relationship between LSECtin and mismatch repair gene. (**F**) Relationship between LSECtin and immune checkpoints. The caption for the red font is COAD. * *p* < 0.05, ** *p* < 0.01, *** *p* < 0.001.

**Figure 7 cancers-14-05300-f007:**
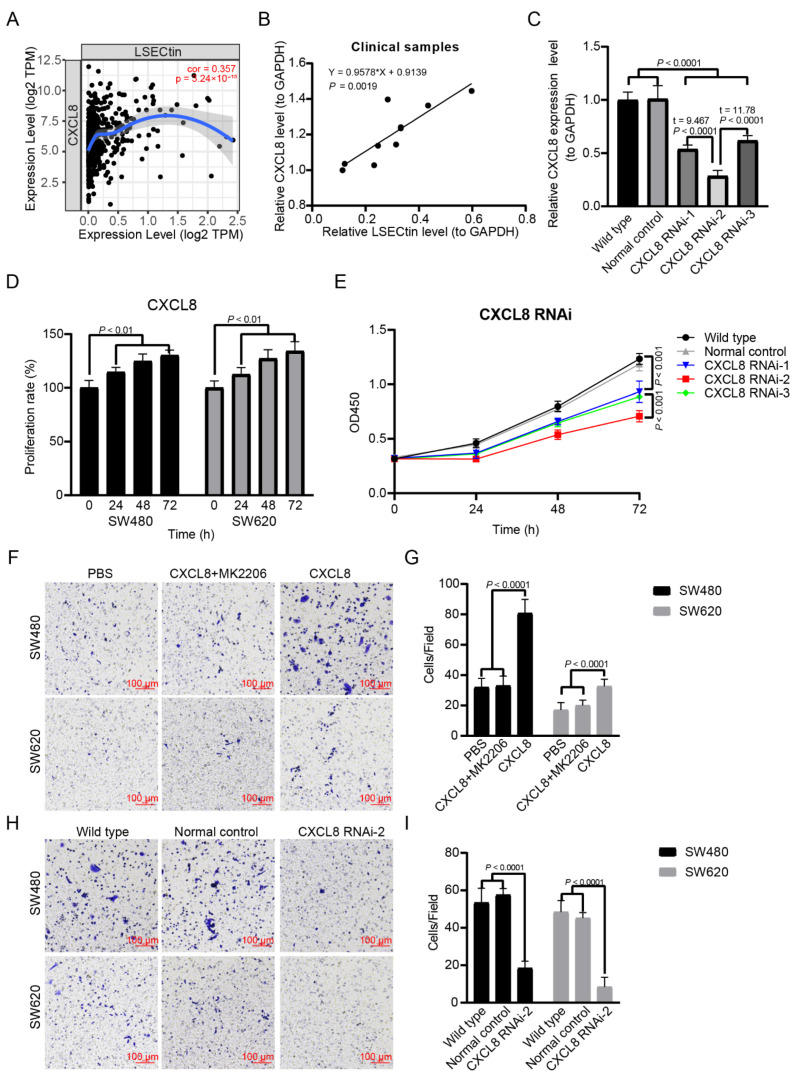
CXCL8 expression is positively correlated with LSECtin and CXCL8 signal is involved in colon cancer proliferation and invasion. (**A**) Correlation between CXCL8 and LSECtin expression (TIMER). (**B**) Correlation between CXCL8 and LSECtin expression (clinical samples). (**C**) Screening of CXCL8 interfering RNA. (**D**) MTT assay was used to detect the effect of CXCL8 (100 ng/mL) on the proliferation of colon cancer cells. (**E**) MTT assay was used to detect the effect of CXCL8 siRNA on the proliferation of colon cancer cells. (**F**,**G**) Transwell assay was used to detect the effect of CXCL8 (100 ng/mL) on the invasion ability of colon cancer cells. (**H**,**I**) Transwell assay was used to detect the effect of CXCL8 siRNA on the invasion ability of colon cancer cells.

**Figure 8 cancers-14-05300-f008:**
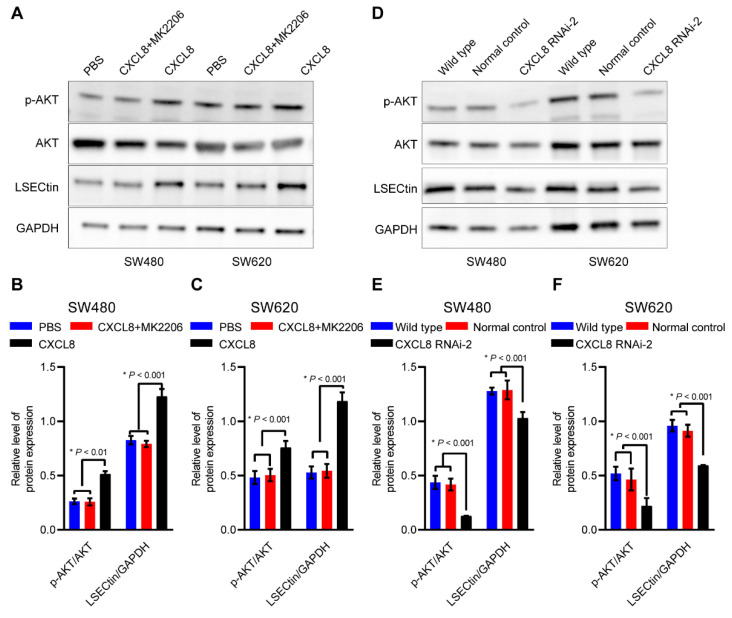
AKT/LSECtin is the downstream of CXCL8 in colon cancer. (**A**) Western blot was used to detect the expression of p-AKT, AKT and LSECtin in colon cancer cells after pretreated with CXCL8 (100 ng/mL). (**B**,**C**) Semi-quantitative analysis results of (**A**). (**D**) Western blot was used to detect the expression of p-AKT, AKT and LSECtin in colon cancer cells after pretreated with CXCL8 siRNA. (**E**,**F**) Semi-quantitative analysis results of (**D**). The uncropped blots are shown in Appendix A.

**Table 1 cancers-14-05300-t001:** Primers used in the current study.

Genes	Reverse	Forward
CXCL8	CACTGCGCCAACACAGAAAT	GCCCTCTTCAAAAACTTCTCCAC
CLEC4G(LSECtin)	ATCTGGGCAAGGTTCAGGGCTA	GCAGCATCATGACACAGTTCTCG
GAPDH	GGAGCGAGATCCCTCCAAAAT	GGCTGTTGTCATACTTCTCATGG

## Data Availability

The data presented in this study are available in this article.

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
