# Peer review of "CXCL8 Up-Regulated LSECtin through AKT Signal and Correlates with the Immune Microenvironment Modulation in Colon Cancer"

_cancers, 2022, doi:10.3390/cancers14215300_

Round 1
Reviewer 1 Report
This manuscript by Fang and colleagues seek to determine the role of CXCL8 and LSECtin in colon cancer. In their previous paper the authors have shown that CXCL8 promotes the liver metastasis of colon cancer through different mechanisms. Here the authors show that CXCL8 can promote the expression of LSECtin in colon cancer cell lines. Overall this is a well designed study that introduces a potential target in colon cancer. Some questions remain:
1) The authors show that CXCL8 treatment promotes the expression of LSECtin. If that's the case, why in colon cancer tissues, the expression of CXCL8 was higher, but LSECtin was lower than that in normal mucosa? Should both their expression higher in cancer tissues? Additional colon cancer dataset may be included to carefully address this question.
2) While the authors show that the higher expression of CXCL8 and LSECtin is positively related to ICIs efficacy markers, the authors also show the positive correlation of CXCL8 with immune activation, which can be used to predict a positive response to ICIs. These results look confusing and controversial.
3) To better characterize the regulation of CXCL8/LSECtin on colon cancer proliferation/invasion, the authors may perform rescue experiments by treating CXCL8-treated cells with LSECtin RNAi or adding recombinant LSECtin to CXCL8i-treated cells.
4) The resolution for F2b&c, F5b&c are very low. The authors should replace these panels with higher resolution images.
5) A reference error on page 13, line 360.
Author Response
Dear reviewer,
Thank you for your constructive comments on our manuscript. We have revised the manuscript accordingly. Detailed responses are given below point by point.
This manuscript by Fang and colleagues seek to determine the role of CXCL8 and LSECtin in colon cancer. In their previous paper the authors have shown that CXCL8 promotes the liver metastasis of colon cancer through different mechanisms. Here the authors show that CXCL8 can promote the expression of LSECtin in colon cancer cell lines. Overall this is a well designed study that introduces a potential target in colon cancer. Some questions remain:
Point 1: The authors show that CXCL8 treatment promotes the expression of LSECtin. If that's the case, why in colon cancer tissues, the expression of CXCL8 was higher, but LSECtin was lower than that in normal mucosa? Should both their expression higher in cancer tissues? Additional colon cancer dataset may be included to carefully address this question.
Response 1: Thank you for your comments. We have also considered this problem. Results from TCGA and GSE110225 both showed that the expression of CXCL8 was higher, but LSECtin was lower in colon cancer tissues than that in normal mucosa. Correlation analysis suggest that the expression of CXCL8 is positively correlated with LSECtin in cancer tissues. According to the general understanding,both of CXCL8 and LSECtin expression should be higher in cancer tissues. However, the truth is not this. Correlation analysis results indirectly suggests that there may be a positive regulatory relationship between CXCL8 and LSECtin in cancer tissues. We have verified these results through clinical sample detection. In addition, our cell experiment also confirmed that CXCL8 positively regulates the expression of LSECtin. Also, it has been reported that CDK12 in tumor tissues was significantly higher than that in non-tumor tissues (4.04±3.25 vs. 1.56±0.84, P=0.009). Expression of CCL21 was not different between tumor and non-tumor tissues (6.56±4.27 vs. 4.65±2.75, P=0.140). But, positive correlation was found between CDK12 and CCL21 in gastric cancer[1]. The complexity of tumor microenvironment may be an important reason for the above phenomena. The large difference of gene expression regulation mechanism between cancer tissue and normal tissue may be the other reason. We also analyzed other data sets, and most of the results suggested that the expression of CXCL8 was higher, but LSECtin was lower in colon cancer tissues than that in normal mucosa. These results were similar to our results in paper. In GSE24551 data, we can find that the expression of CXCL8 is significantly positively correlated with LSECtin in cancer tissues (there are 174 samples in this data set). From the above evidence, it can be suggested that the results in our paper are reliable.
The verification results are shown in the following figure:
Figure 1. Verification results from GSE113513, GSE184093 and GSE24551.
[1] Ji J, Zhou C, Wu J, Cai Q, Shi M, Zhang H, Yu Y, Zhu Z, Zhang J. Expression pattern of CDK12 protein in gastric cancer and its positive correlation with CD8+ cell density and CCL12 expression. Int J Med Sci. 2019 Aug 6;16(8):1142-1148. doi: 10.7150/ijms.34541. PMID: 31523177; PMCID: PMC6743279.
Point 2:While the authors show that the higher expression of CXCL8 and LSECtin is positively related to ICIs efficacy markers, the authors also show the positive correlation of CXCL8 with immune activation, which can be used to predict a positive response to ICIs. These results look confusing and controversial.
Response 2: This is a good question, and we also made some explanations in the discussion. Our data suggest that the high expression of CXCL8 in colon cancer is positively correlated with the therapeutic markers of ICIs, and the infiltration of immune cells. So, theoretically, patients with high CXCL8 expression should be sensitive to ICIs treatment. However, current clinical data suggest that patients with high CXCL8 expression are less sensitive to ICIs treatment. Our results also suggest that the expression of CXCL8 in colon cancer is positively correlated with the overexpression of more than 40 immune checkpoints. Therefore, the increased expression of immune checkpoints may be the main reason in weakening the sensitivity to ICIs treatment in patients with CXCL8 overexpression. This speculation still needs further research and verification.
Point 3:To better characterize the regulation of CXCL8/LSECtin on colon cancer proliferation/invasion, the authors may perform rescue experiments by treating CXCL8-treated cells with LSECtin RNAi or adding recombinant LSECtin to CXCL8i-treated cells.
Response 3: Thank you very much for your suggestion. This rescue experiment is necessary to further clarify the relationship between CXCL8 and LSECtin. Our team will construct stably shCXCL8 cells through shRNA, clarify the relationship between CXCL8 and LSECtin again through LSECtin shRNA, determine the impact of shCXCL8 cells on T cell function, and carry out animal research. We have done a lot of preliminary work on CXCL8 and colon cancer, and we will continue to track and report the latest research results. Thank you again for your suggestions, which have helped us a lot for future work.
Point 4: The resolution for F2b&c, F5b&c are very low. The authors should replace these panels with higher resolution images.
Response 4: Because word documents compress the image again, the resolution is low. We have changed it to a high-resolution figure in the revised manuscript.
Point 5: A reference error on page 13, line 360.
Response 5: We are very sorry that we have made such mistakes. We have revised and carefully checked the references in this paper.

Reviewer 2 Report
Basic reporting and Comments
In this manuscript authors have Identified that CXCL8 upregulates the LSECtin through Akt pathway in colon cancer and thereby contribute to the immune checkpoint inhibitors resistance.
1. English needs to be checked throughout the manuscript. Summary and abstract needs to be rephrased. Use of the word “can” is not appropriate while describing your conclusions and results. Also, the sentences are not having proper breaks, they are very lengthy.
2. Study lacks novelty somehow, as its correlative in a sense that expression levels of CXCL8 from patient samples already recorded in the databases is compared with new patient samples. It is kind of validation of already known information with new samples.
3. How does LSECtin modulates CXCL8 in your models? There are some preliminary reports suggesting LSECtin downregulates the CXCL8.
https://www.jimmunol.org/content/206/1_Supplement/25.06
4. It is always good to do knockdown experiments with shRNA, which is more stable and targeted.
5. Typo in line 199, for the gene expression formula.
6. It is good to label expression levels in all the RT-PCR data rather than just saying levels.
7. Font size is too small to read in Figure 2, 3, 5 and 6.
8. Whereever you are mentioning the word “same results” it has to be replaced with “similar results”. It is more appropriate.
9. As you showed CXCL8 knockdown decreases the invasion ability of colon cancer cells, why not check some of the expression of the EMT markers (Vimentin, slug, snail and twist) in these knockdown cells as well?
10. Please label “transwell assay” in line 309 and 310 instead of transwell. It is more appropriate.
11. GAPDH and Akt loading control does not seem to be equal in Fig 8.
12. When study focuses on finding the role of CXCL8 and LSECtin in the immune micrenvironment, some more coculture experiments need to be performed. Like either incubating the supernatant of CXCL8 transfected cells with the T cells to check for the function of T cells, as to how they are modulated with the LSECtin upregulation. Majority of the study is corelative, it is good to do more experiments in cell line or mice to prove the concept.
13. I do not see much significance of the Figure 7, as there are many reports which already state that CXCL8 is overexpressed in colon cancre patients and it promotes EMT, that concludes that if we knockdown CXCL8, it will lead to inihibition of colon cancer cell growth. Also, it is known that CXCL8 induces EMT through AKt pathway.
14. Many relevant citations are missing throughout the manuscript. Need revision on that.
Author Response
Dear reviewer,
Thank you for your constructive comments on our manuscript. We have revised the manuscript accordingly. Detailed responses are given below point by point.
Basic reporting and Comments
In this manuscript authors have Identified that CXCL8 upregulates the LSECtin through Akt pathway in colon cancer and thereby contribute to the immune checkpoint inhibitors resistance.
Point 1: English needs to be checked throughout the manuscript. Summary and abstract needs to be rephrased. Use of the word “can” is not appropriate while describing your conclusions and results. Also, the sentences are not having proper breaks, they are very lengthy.
Response 1: We have reviewed our manuscript carefully, mistakes were revised as can be seen in the new “marked-up” manuscript.
Point 2: Study lacks novelty somehow, as its correlative in a sense that expression levels of CXCL8 from patient samples already recorded in the databases is compared with new patient samples. It is kind of validation of already known information with new samples.
Response 2: Our study completed the analysis of CXCL8 in colon cancer from existing public data, we know that some results related to CXCL8 have been reported before. However, the focus of this study is to find that the expression of CXCL8 is significantly positively correlated with LSECtin in colon cancer, and our cell experiment also confirmed that CXCL8 positively regulates the expression of LSECtin. These results have not been reported yet. The results of this study can also partly explain why the patient with high CXCL8 expression is less sensitive to ICIs treatment. Therefore, our research have certain novelty and scientific value. It has certain theoretical reference value to further explore the primary drug resistance of ICIs in colon cancer.
Point 3: How does LSECtin modulates CXCL8 in your models? There are some preliminary reports suggesting LSECtin downregulates the CXCL8.
https://www.jimmunol.org/content/206/1_Supplement/25.06
Response 3: This is a question worth thinking about. Our team ignored this question. We haven't retrieved this report you provided. The results suggest that LSECtin negatively regulates the expression of CXCL8 in activated T cells. Because our early research object is CXCL8, most of our research is more based on CXCL8. In the future, we can also explore the function of LSECtin in colorectal cancer. Thank you again for your suggestions.
Point 4: It is always good to do knockdown experiments with shRNA, which is more stable and targeted.
Response 4: Thank you for your constructive comments. Our team will use this technology to construct cells stably knockout CXCL8, clarify the relationship between CXCL8 and LSECtin expression, analyze the impact of the shCXCL8 cells on T cell function, and carry out animal studies. We have done a lot of preliminary work on CXCL8 and colorectal cancer. We will report the relevant results in a future report. Thank you again for your suggestions, which have helped us a lot for our future work.
Point 5: Typo in line 199, for the gene expression formula.
Response 5: We have revised them. Thank you!
Point 6: It is good to label expression levels in all the RT-PCR data rather than just saying levels.
Response 6: Thank you very much for your advice. We have changed all of the label.
Point 7: Font size is too small to read in Figure 2, 3, 5 and 6.
Response 7: We have changed then to high-resolution figures in the revised manuscript. The font size is also increased.
Point 8: Whereever you are mentioning the word “same results” it has to be replaced with “similar results”. It is more appropriate.
Response 8: Thank you very much for your advice. There are some imprecise expressions in our manuscript. We have reviewed our manuscript carefully and revised them.
Point 9: As you showed CXCL8 knockdown decreases the invasion ability of colon cancer cells, why not check some of the expression of the EMT markers (Vimentin, slug, snail and twist) in these knockdown cells as well?
Response 9: EMT is an important mechanism of tumor cell invasion and metastasis. Our previous research has confirmed that CXCL8 can induce EMT in Lovo [1] and partial EMT in SW480 [2]. Therefore, in this study, we did not carry out relevant detection after CXCL8 was knocked out.
[1] Shen T, Yang Z, Cheng X, Xiao Y, Yu K, Cai X, Xia C, Li Y. CXCL8 induces epithelial-mesenchymal transition in colon cancer cells via the PI3K/Akt/NF-κB signaling pathway. Oncol Rep. 2017 Apr;37(4):2095-2100. doi: 10.3892/or.2017.5453. Epub 2017 Feb 14. PMID: 28259918.
[2] Cheng, X.; Li, Y.; Tan, J.; Sun, B.; Xiao, Y.; Fang, X.; Zhang, X.; Li, Q.; Dong, J.; Li, M.; et al. CCL20 and CXCL8 synergize to promote progression and poor survival outcome in patients with colorectal cancer by collaborative induction of the epithelial-mesenchymal transition. Cancer letters 2014, 348, 77-87, doi:10.1016/j.canlet.2014.03.008.
Point 10: Please label “transwell assay” in line 309 and 310 instead of transwell. It is more appropriate.
Response 10: We have revised them. Thank you!
Point 11: GAPDH and Akt loading control does not seem to be equal in Fig 8.
Response 11: The three lanes on the left and the three lanes on the right are SW480 and SW620 cell lines, respectively. Therefore, there is a certain difference in the sample loading on both sides, which also causes the inequality between the left three lanes and the right three lanes. But it does not affect our interpretation of the results. At the same time, the semi-quantitative analysis of the western blots was conducted using Image J software to make the results more clear.
Point 12: When study focuses on finding the role of CXCL8 and LSECtin in the immune micrenvironment, some more coculture experiments need to be performed. Like either incubating the supernatant of CXCL8 transfected cells with the T cells to check for the function of T cells, as to how they are modulated with the LSECtin upregulation. Majority of the study is corelative, it is good to do more experiments in cell line or mice to prove the concept.
Response 12: Thank you for your constructive comments. Our team will use shRNA to construct cells stably knockout CXCL8, analyze its impact on T cell function, and carry out animal research. We will report the relevant results in a future report.
Point 13: I do not see much significance of the Figure 7, as there are many reports which already state that CXCL8 is overexpressed in colon cancre patients and it promotes EMT, that concludes that if we knockdown CXCL8, it will lead to inihibition of colon cancer cell growth. Also, it is known that CXCL8 induces EMT through AKt pathway.
Response 13: .The role of CXCL8 in the invasion, metastasis and proliferation of colorectal cancer has been widely reported, and we also have similar research [1,2]. The data involved in Figure 7 indicated that our experiment to knock down CXCL8 is successful. Therefore, figure 7 has certain reference significance for the discussion of this paper.
[1] Cheng, X.; Li, Y.; Tan, J.; Sun, B.; Xiao, Y.; Fang, X.; Zhang, X.; Li, Q.; Dong, J.; Li, M.; et al. CCL20 and CXCL8 synergize to promote progression and poor survival outcome in patients with colorectal cancer by collaborative induction of the epithelial-mesenchymal transition. Cancer letters 2014, 348, 77-87, doi:10.1016/j.canlet.2014.03.008.
[2] Xiao, Y.; Yang, Z.; Cheng, X.; Fang, X.; Shen, T.; Xia, C.; Liu, P.; Qian, H.; Sun, B.; Yin, Z.; et al. CXCL8, overexpressed in colorectal cancer, enhances the resistance of colorectal cancer cells to anoikis. Cancer letters 2015, 361, 22-32, doi:10.1016/j.canlet.2015.02.021.
Point 14: Many relevant citations are missing throughout the manuscript. Need revision on that.
Response 14: We are very sorry that we have made some mistakes in citations. We have revised and carefully checked the references in this paper.

Round 2
Reviewer 1 Report
The authors have addressed my concerns.